



# A reduced-complexity model of fluvial inundation with a sub-grid representation of floodplain topography evaluated for England, United Kingdom

Simon J. Dadson[a,b], Eleanor Blyth[a], Douglas Clark[a], Helen Davies[a], Richard Ellis[a], Huw Lewis[c], Toby Marthews[a], and Ponnambalam Rameshwaran[a]

[a]UK Centre for Ecology and Hydrology, Maclean Building, Crowmarsh Gifford, Wallingford, OX10 8BB, United Kingdom.
[b]School of Geography and the Environment, University of Oxford, OX1 3QY, United Kingdom.
[c]Met Office, FitzRoy Road, Exeter EX1 3PB, United Kingdom.

**Correspondence:** Simon J. Dadson (sjdad@ceh.ac.uk)

**Abstract.** Timely predictions of fluvial flooding are important for national and regional planning and real-time flood response. Several new computational techniques have emerged in the past decade for making rapid fluvial flood inundation predictions at time and space scales relevant to early warning, although their efficient use is often constrained by the trade-off between model complexity, topographic fidelity and scale. Here we apply a simplified approach to large-area fluvial flood inundation modelling which combines a solution to the inertial form of the shallow water equations at 1 km horizontal resolution, with two alternative representations of sub-grid floodplain topography. One of these uses a fitted sub-grid probability distribution, the other a quantile-based representation of the floodplain. We evaluate the model's performance when used to simulate the 0.01 Annual Exceedance Probability (AEP; '100-year') flood and compare the results with published benchmark data for England. The quantile-based method accurately predicts flood inundation in 86% of locations, with a domain-wide hit rate of 95% and false alarm rate of 10%. These performance measures compare with a hit rate of 71%, and false alarm rate of 9% for the simpler, but faster, distribution-based method. We suggest that these approaches are suitable for rapid, wide-area flood forecasting and climate change impact assessment.

## 1 Introduction

Flooding is a costly and damaging natural hazard which is projected to increase under climate change. Society benefits from access to timely and accurate flood forecasts and risk-based hazard maps (Teng et al., 2017; Ward et al., 2015). Recent advances in large-scale fluvial flood risk modelling have made it possible to conduct wide-area analyses of fluvial flooding across large geographical extents (Bradbrook, 2006; Dottori et al., 2017; Pappenberger et al., 2012; Wing et al., 2017). These advances are attributed in part to recently-developed algorithms which solve for lateral spreading of floodwaters using the shallow water equations, albeit with simplifying assumptions and constraints on model solution properties imposed for computational stability (Bates and De Roo, 2000; Bates et al., 2010). Well-tested techniques have emerged which implement adaptive time-step control (e.g., de Almeida and Bates, 2013) and which employ solution meshes with variable spatial resolutions (e.g., Liang and





Borthwick, 2009). Moreover, new methods to represent sub-grid variability have brought benefits in cases where fine resolution topography can be represented statistically within a coarser resolution model grid box (e.g., Dadson et al., 2010; Neal et al., 2012a; Yamazaki et al., 2011). Alongside these computational advances, high-resolution topographic datasets are now available globally and regionally. Even though these datasets do not explicitly resolve some important sub-grid features including flood

defences, they have considerably improved our knowledge of the underpinning topographic boundary conditions (e.g., Döll and Lehner, 2002; Morris and Flavin, 1990; Yamazaki et al., 2019). These challenges have been tackled to compute static hazard maps for a range of annual exceedance probabilities (AEPs; e.g., Lamb et al., 2009; Wing et al., 2017), and to provide local area, real-time predictions (e.g., Xia et al., 2019; Yu et al., 2016).

Notwithstanding these innovations, the application of complex shallow water flow routing codes over large domains is still

computationally intensive. There therefore remains a need for algorithms that offer computational efficiency whilst maintaining predictive accuracy (Salas et al., 2018; Schumann et al., 2013). After widespread flooding in the United Kingdom in 2007, an independent review recommended that a capability for distributed flood prediction be developed at $\sim$ 1 km resolution, in order to improve understanding of flood risk across a national-scale domain (Pitt, 2008, p.53). After further extreme flooding in 2015–2016, the National Flood Resilience Review called for "a more integrated flood risk modelling approach to allow

simulations to be run which link meteorology, hydrology and flooding across England" (Cabinet Office, 2016, p.26). To address these needs, simplified approaches have been applied to reduce computation time further, although their accuracy is yet to be evaluated fully (Afshari et al., 2018; Hall et al., 2003; Johnson et al., 2019; Nobre et al., 2016).

Our purpose in this study is therefore to apply a computationally-efficient simulation model which retains the physical principles underpinned by the shallow water equations, but which uses a sub-grid parametrisation to represent relevant information

on the sub-grid floodplain. This approach is not intended to replace traditional site-specific flood models, for which best practice remains measurement of channel cross sections combined with detailed topographic surface models in which details of defences and other flood management interventions can be delineated. By contrast, the method introduced here is designed for use in circumstances where the requirement for computational efficiency precludes a more complex approach, e.g., in wide-area flood-forecasting and prediction of climate change impacts. In practice, the model discussed in this paper is intended to be

incorporated into the JULES land-surface model (Best et al., 2011) via the Hydro-JULES interface framework (NERC, 2018). The goal of this study is therefore an offline evaluation of this model component against benchmark data available for England, United Kingdom. In this paper we test the following specific hypotheses: (i) that a 1 km model with sub-grid floodplain can perform accurately compared with benchmark data; and (ii) that the additional complexity of a quantile-based approach improves model performance when compared with a simpler distribution-based method.

## 2  Data and methods

### 2.1  Model description and evaluation domain

The approach taken here combines two model components. The first calculates the lateral distribution of floodplain flows between grid cells, given storage in each cell, based on the inertial form of the shallow water equations proposed by de Almeida





and Bates (2013). The second component accounts for the sub-grid floodplain elevation distribution. For this second, sub-grid component we compare two approaches. The first uses a log-normal approximation to the floodplain topography based on earlier work by Dadson et al. (2010); the second adopts a quantile-based discretisation of the floodplain, similar to that used in CaMa-Flood (Yamazaki et al., 2011). The model's state variables are arranged so that each model grid box contains a volume

of water in its prognostic open water store (Figure 1; and see Table A1 for notation). From this volume, the model calculates a corresponding depth of water, $h$, and area inundated, $A_{fl}$, conserving mass within each grid box (see Section 2.4). For the purposes of the current test we select our model domain to be the area encompassed by the Environment Agency's Indicative Floodplain map for England (Figure 2), which we represent at 1 km horizontal resolution.

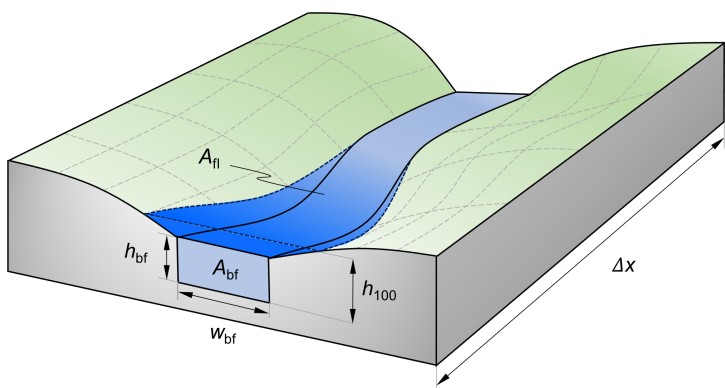

**Figure 1.** Schematic diagram defining modelled quantities: $A_{bf}$, channel cross-sectional area (at bankfull depth); $w_{bf}$, grid box mean channel width; $h_{bf}$, bankfull depth; $h_{100}$, depth at AEP 0.01 flood; $\Delta x$, horizontal grid box length; $A_{fl}$, inundated fraction.

## 2.2 Flood depth estimation

Several methods can be used to compute flood depths at the scale of this study. The first converts flows estimated statistically for a given recurrence interval everywhere along the channel network to depths, using a flow resistance equation such as Manning's. Here we adopt an alternative technique in which flood depths are estimated directly, based on measured attributes at each location in the domain. The justification for this approach is to maintain close correspondence with the method used to compute the benchmark dataset (Morris and Flavin, 1990). This decision does not preclude the subsequent use of flow-derived

flood depths, for example to perform event-based analyses, although clearly to do so would introduce additional uncertainty.

Using a database of extreme river depths compiled explicitly for the purpose, we compute 0.01 AEP flood depths, $h_{100}$, above bankfull across the modelled domain (Naden and McCartney, 1991). It is acknowledged that the 0.01 AEP event is highly unlikely to occur everywhere simultaneously, but for comparability with the widely-used Environment Agency benchmark data we maintain this approach. A multiple linear regression was employed with log-transformed inputs to estimate $h_{100}$ as

a function of three variables: (i) catchment area, $A_{cat}$, computed from the 50 m UK Institute of Hydrology Digital Terrain Model (IHDTM; Figure 2a; Morris and Flavin, 1990), (ii) standardised annual average rainfall, (SAAR; Figure 2b; Hollis





et al., 2018), and (iii) standardised percentage runoff, SPRHOST, computed from Hydrology of Soil Types classes (HOST; Figure 2c; Boorman et al., 1995). This method is modified from the procedure described by Naden and McCartney (1991) in which we use HOST instead of the Flood Studies Report Winter Rainfall Acceptance Potential (WRAP) classes, owing to HOST's more detailed consideration of subsurface hydrology (Figure 2c).

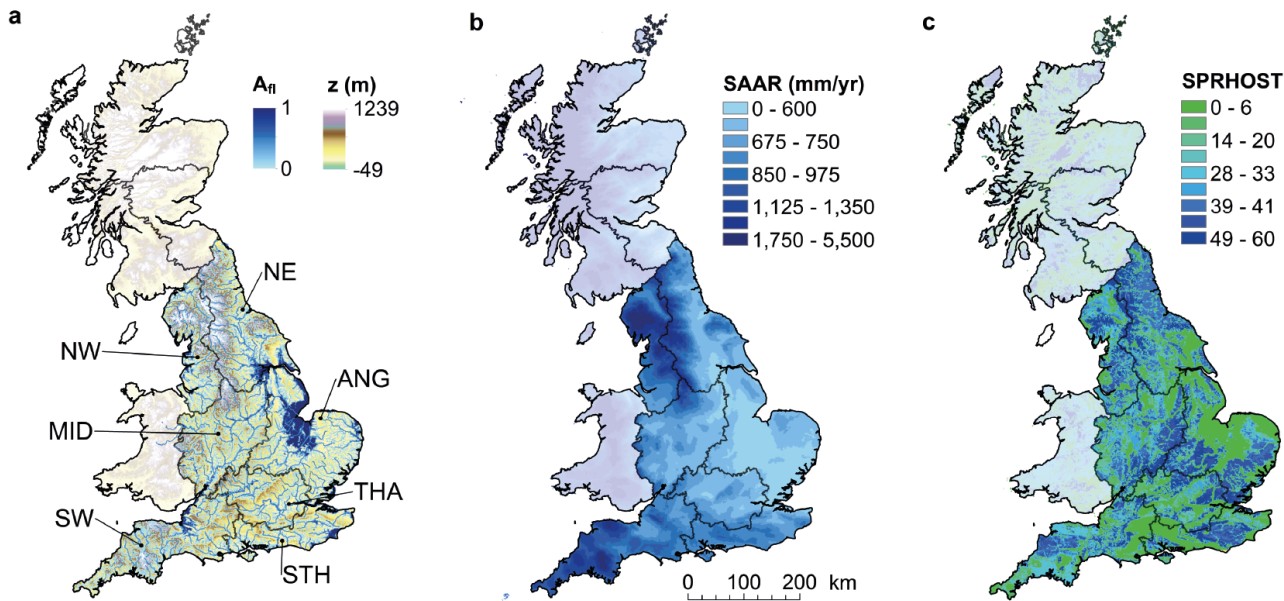

**Figure 2.** Supporting datasets a, Environment Agency Indicative Floodplain map shown over 50 m IHDTM hydrologically-corrected terrain model. Reporting regions are NE, North East; ANG, East Anglia; THA, Thames; STH, South East; SW, South West; MID, Midland; NW, North West; b, Standardised Annual Average Rainfall (SAAR, mm; blue shading); c, Standardised percentage runoff from Hydrology of Soil Types dataset (SPRHOST, %; green-blue shading). Areas of the British mainland outside England are shown masked because validation data are available only for England, for which subsequent plots are shown. The standard UK National Grid transverse Mercator projection is used together with the Ordnance Survey of Great Britain (OSGB) datum with origin at 2°W, 49°N (Ordnance Survey, 2018).

5    These variables predicted bankfull depth with statistical significance at the 0.05 level (with the following standard regression diagnostics calculated: multiple $R^2 = 0.64$, $F(3, 30) = 17.6$; $p = 8.8 \times 10^{-7}$; see Table 1). The resulting estimator of $h_{bf}$ is given in Equation 1, and plotted in Figure 3 using the fitted values: $a = 1.93 \times 10^{-3}$, $b = 0.29$, $c = 0.47$, and $d = 0.64$.

$$\hat{h}_{bf} = a(\text{AREA})^b(\text{SAAR})^c(\text{SPRHOST})^d \tag{1}$$





**Table 1.** Analysis of variance statistics for regression model fitted using Equation 1. Stars indicate significance; Df, degrees of freedom.

|  | **Df** | **Sum Sq** | **Mean Sq** | **F Value** | **Pr(>F)** |
|---|---|---|---|---|---|
| $\ln(A_{cat})$ | 1 | 4.50 | 4.50 | 32.92 | $2.91 \times 10^{-6}$ *** |
| $\ln(\text{SAAR})$ | 1 | 1.43 | 1.43 | 10.45 | $2.97 \times 10^{-3}$ ** |
| $\ln(\text{SPRHOST})$ | 1 | 1.29 | 1.29 | 9.44 | $4.49 \times 10^{-3}$ ** |
| Residuals | 30 | 4.10 | 0.14 |  |  |

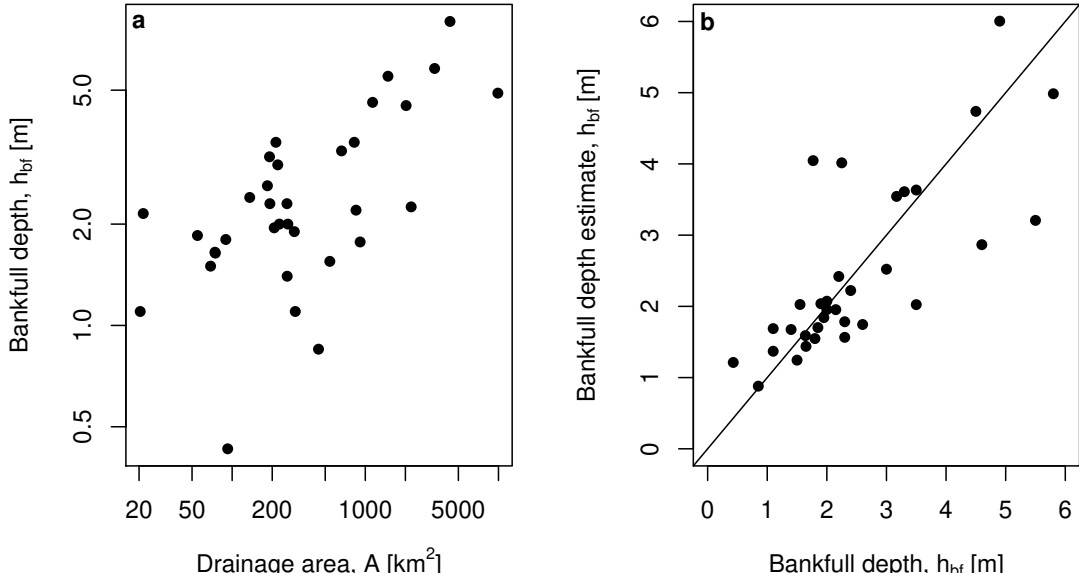

**Figure 3.** Bankfull depth relations. a, relation between bankfull depth, $h_{bf}$, and drainage area, $A_{cat}$; b, residuals from the regression relation for $n = 33$ measurements reported by Naden and McCartney (1991). The solid line shows the 1:1 relation.

Having computed $h_{bf}$ (at bankfull) we convert this depth to an equivalent 0.01 AEP flood height $h_{100}$ using the following scaling relation designed for direct estimation of flood depth:

$$\hat{h}_{100} = m\hat{h}_{bf}^{p} \tag{2}$$

where m = 6.75 and p = 0.2 are calculated from values given by Naden and McCartney (1991).





## 2.3 Lateral spreading of floodplain flows

Our simulation of lateral distribution of floodplain flows uses the equation of de Almeida and Bates (2013). This formulation simplifies the solution of the depth-integrated form of the Navier-Stokes equations by neglecting convective acceleration – assuming gradually varying, sub-critical flow – to give the following continuity and simplified momentum equations (Equations 3 and 4, respectively):

$$\frac{\partial h}{\partial t} + \frac{\partial q_i}{\partial x_i} = 0, \tag{3}$$

and

$$\frac{\partial q_i}{\partial t} + gh\frac{\partial q_i(h+z)}{\partial x_i} + \frac{gn^2 q_i^2}{R^{4/3}h} = 0, \tag{4}$$

where $t$ is time (s), $x_i$ are coordinates in space (m), $q_i$ is the component of flow of water per unit channel width in the direction $x_i$ (in m$^2$ s$^{-1}$), $g$ is the gravitational acceleration (m s$^{-2}$), $h$ is water surface elevation (m), $z$ is topographic elevation of the channel bed (m), $n$ is the Gauckler-Manning roughness coefficient (s m$^{-1/3}$), and $R$ is hydraulic radius (m). An explicit form of the discretised update equation for flow per unit channel width can be obtained from Equation (4) by assuming that for shallow flows $R$ can be approximated by $h$ and by using the finite difference scheme of de Almeida and Bates (2013),

$$q_i^{t+\Delta t} = \frac{q_i^t - gh_t\Delta t\frac{\partial(h_t+z)}{\partial x_i}}{1 + g\Delta t n^2|q_i^t|/h_t^{7/3}}. \tag{5}$$

The stability of this solution depends on the Courant-Friedrichs-Lewy condition:

$$\Delta t = \alpha\frac{\Delta x}{\sqrt{gh_{max}}}, \tag{6}$$

where $h_{max}$ is defined as the maximum depth of water within the domain of the model (m) and $\alpha$ is a dimensionless factor between zero and unity. In the present study we use an adaptive time-step solver based on Equations 3–6 with $\alpha = 0.7$ to improve stability (de Almeida and Bates, 2013, p.4836).

In addition to the initial state of the model, $h(x_i)$, two other pieces of information are required: a dataset on channel width, $w_{bf}$, which is needed to calculate flow per unit width, $q_i$; and information on the Gauckler-Manning roughness coefficient, $n$. We calculate channel width using the same dataset used in Table 1. However, in the case of channel width only upslope area





and soils data are used because the effect of rainfall is not statistically significant. These variables predicted bankfull width with statistical significance at the 0.05 level (Multiple $R^2$ = 0.70, $F(2,27)$ = 32.4; $p = 6.7 \times 10^{-8}$; see Table 2). Note that fewer locations in the dataset had a well-defined bankfull width than had a recorded depth, hence the slightly lower number of observations used in this analysis. The resulting estimator of bankfull width is given by Equation 7 and plotted in Figure 4

using the fitted values: $g$= 0.157, $j$ = 0.48, and $k$ = 0.51.

$$\hat{w}_{bf} = g(\mathrm{AREA})^{j}(\mathrm{SPRHOST})^{k}. \tag{7}$$

**Table 2.** Analysis of variance statistics for regression model fitted using Equation 7. Stars indicate significance; Df, degrees of freedom.

|  | Df | Sum Sq | Mean Sq | F Value | Pr(>F) |
|---|---|---|---|---|---|
| $\ln(A_{cat})$ | 1 | 11.57 | 11.57 | 60.56 | $2.28 \times 10^{-8}$ *** |
| $\ln(\mathrm{SPRHOST})$ | 1 | 0.81 | 0.81 | 4.26 | 0.0487 * |
| Residuals | 27 | 5.16 | 0.19 |  |  |

    Gridded channel roughness estimates were derived from a database of river cross-section surveys (Environment Agency, 2020). For each surveyed river reach, a composite roughness estimate was obtained by weighting roughness measurements from Fisher and Dawson (2003) according to surveyed fractional size classes of bed material. Weights were chosen to respect

the condition that the total resistance to flow across the river cross-section is equal to the sum of the resistance to flow in each of the substrate classes (Chow, 1959). The resulting composite roughness values were interpolated onto the river channel network using natural neighbour interpolation. For locations not on the channel network a standard floodplain roughness value of 0.04 was used. The sensitivity of our results to roughness is evaluated in Section 4.1.

    Here we consider only fluvial inundation -– not groundwater, pluvial or coastal flooding – although the model developed

here is formulated so as to be able to receive rainfall inputs, supply recharge to a groundwater model, and obey a downstream flux or level constraint imposed by a coastal or estuarine boundary condition in the future. It should be noted that, in common with other models typically applied at horizontal discretisation scales greater than ∼10 m, this approach is not designed to resolve flow shocks and is therefore unsuitable for flows with Froude number greater than unity. For the applications under consideration here this is not a restrictive limitation (Neal et al., 2012b).

In order to compute the inundated extent associated with the AEP 0.01 flood, we track the time evolution of maximum inundated area in a transient simulation with a temporal integration defined by the relaxation time, $\tau^*$, of a typical perturbation on the solution mesh, where

$$\tau^* \sim n(SA^*)^{1/2}h^{1/3}. \tag{8}$$

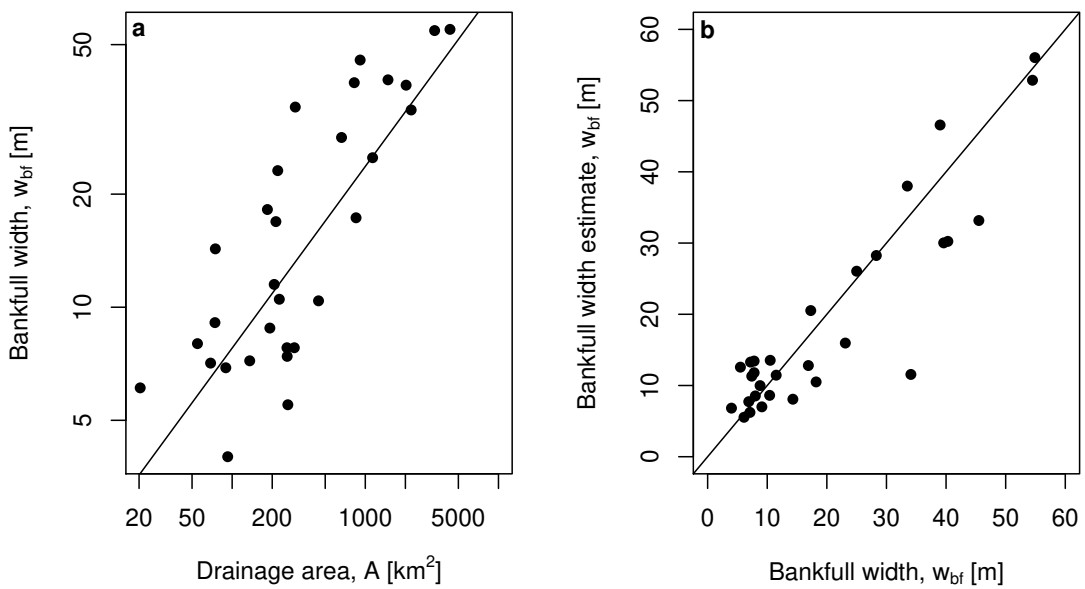

**Figure 4.** Bankfull width relations. a, relation between bankfull width, $w_{bf}$, and drainage area, $A_{cat}$; b, residuals from the regression relation; solid line shows the 1:1 relation.

Here $A_{cat}^*$ is the maximum catchment area within the domain of interest and $c^*$ is a characteristic wave celerity obtained by differentiating the Manning equation:

$$c^* = \frac{\partial q}{\partial h} = 2h^{-1/3}S^{-1/2}/3n. \tag{9}$$

In this definition, $h$ is the typical water elevation (above bankfull) across the domain and $S$ is the typical channel slope. For the
5   domain used here, $A_{cat}^* \sim 2 \times 10^{10}$ m$^2$, $h \sim 1$ m, $S \sim 0.001$ and $n \sim 0.04$, gives an expected equilibration or 'spin-up' time, $\tau^* \sim 3 \times 10^5$ s. The typical runtime to equilibrium for the $\sim 700 \times 10^3$ point British mainland domain shown in Figure 2, using a single core on a standard 3 GHz Intel processor with 16 GB 1.6 GHz DDR3 RAM, is 7–12 minutes.

## 2.4 Sub-grid topographic parametrisation

Having computed the laterally-routed water depth, the inundated area for each grid box is computed with reference to the under-
10   lying topography, subject to the constraint that mass is conserved. Here we use sub-grid topography to calculate a parametrized relation between depth above bankfull and the fraction of the floodplain in the grid box which is inundated (Figure 6). This rela-



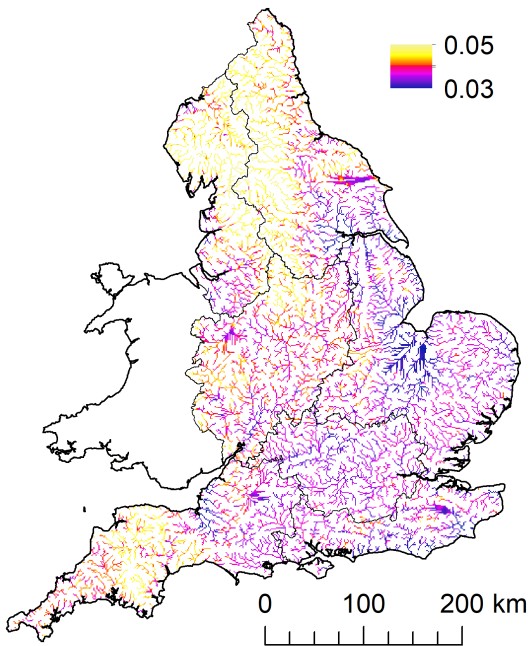

**Figure 5.** Gauckler-Manning roughness coefficient, $n$, estimated from channel bed texture properties. Data: Environment Agency (2020).

tion may take many forms and has been modelled according to a log-normal distribution by Dadson et al. (2010) and piecewise by Yamazaki et al. (2011). The former approach is less computationally demanding; the latter makes fewer assumptions about the nature of floodplain topography but requires that more sub-grid information be retained in supporting datasets. In this study, we compare the two approaches and evaluate their performance and sensitivity to model parameters. In each case, sub-grid

5  elevation data were taken from the IHDTM, which is a hydrologically-corrected digital terrain model with 50 m horizontal resolution, produced from Ordnance Survey data (Morris and Flavin, 1990). In each sub-grid floodplain model, inundated extent is computed as a function of water stored in the grid box. Therefore we make the simplifying assumption that as the grid box drains, its inundated area recedes simultaneously (i.e., sub-grid flood retention time is negligible).

### 2.4.1  Log-normal sub-grid floodplain model

10  To construct the log-normal sub-grid floodplain model, we characterise the sub-grid elevation, $z(x,y)$, above the minimum elevation in that grid box, using a log-normal distribution, such that $\ln(z) \sim N(\mu, \sigma^2)$. Maximum-likelihood estimators are constructed with reference to the sub-grid elevation data such that for each large (1 km) model grid box, $s$, containing $n$ sub-grid elevations, $z_{i,j}$:

$$\hat{\mu}(s) = \frac{\Sigma_{i,j} \ln z_{i,j}}{n} \tag{10}$$





$$\hat{\sigma}^2(s) = \frac{\Sigma_{i,j}\left(\ln z_{i,j} - \hat{\mu}\right)^2}{n} \qquad (11)$$

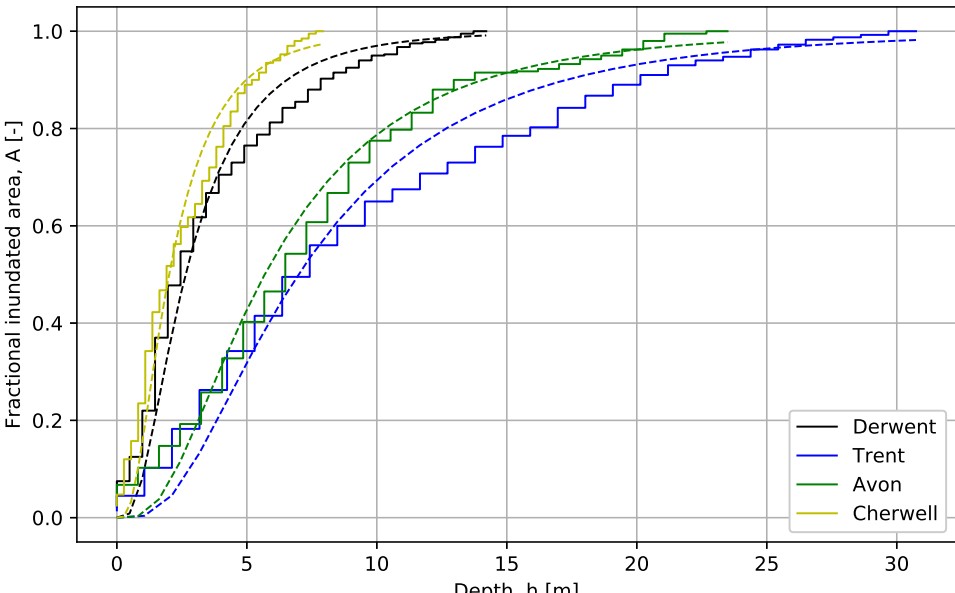

**Figure 6.** Sub-grid topographic properties showing examples of sub-grid topographic cumulative distributions (solid lines) together with fitted sub-grid model (dashed lines). Examples are shown for selected floodplain locations in the Derwent (black, 0.46°W, 54.20°N), Trent (blue, 1.38°W, 52.81°N), Avon (green, 2.24°W, 51.33 °N), and Cherwell catchments (yellow, 1.25°W, 51.79°N) in England, UK.

The validity of the sub-grid topographic approximation is evaluated by considering the root-mean squared error (RMSE) between the approximated elevation profile and the observed one. The median value of this RMSE is 13%. Figure 6 shows example profiles with their fitted equivalents showing close correspondence, albeit with some divergences at depths so large that they are unlikely to be reached in any physically-plausible scenario. Once these parameters have been estimated from sub-grid elevation, the area of the grid box below a given elevation, $A(z^*)$, is obtained using the cumulative distribution function for the log-normal distribution as follows:

$$A(z^*) = \frac{1}{2} + \frac{1}{2}\text{erf}\left[\frac{\ln z^* - \mu}{\sqrt{2}\sigma}\right], \qquad (12)$$





in which $\mathrm{erf}(s)$ is the error function defined by Abramowitz and Stegun (1964) as:

$$\mathrm{erf}(s) = \frac{2}{\sqrt{\pi}} \int_0^s e^{-\xi^2} \, d\xi,$$ (13)

Maps showing values of the parameters $\mu$ and $\sigma$ are plotted in Figure 7 which shows small values corresponding to low, flat terrain and large values in high, rugged terrain, respectively.

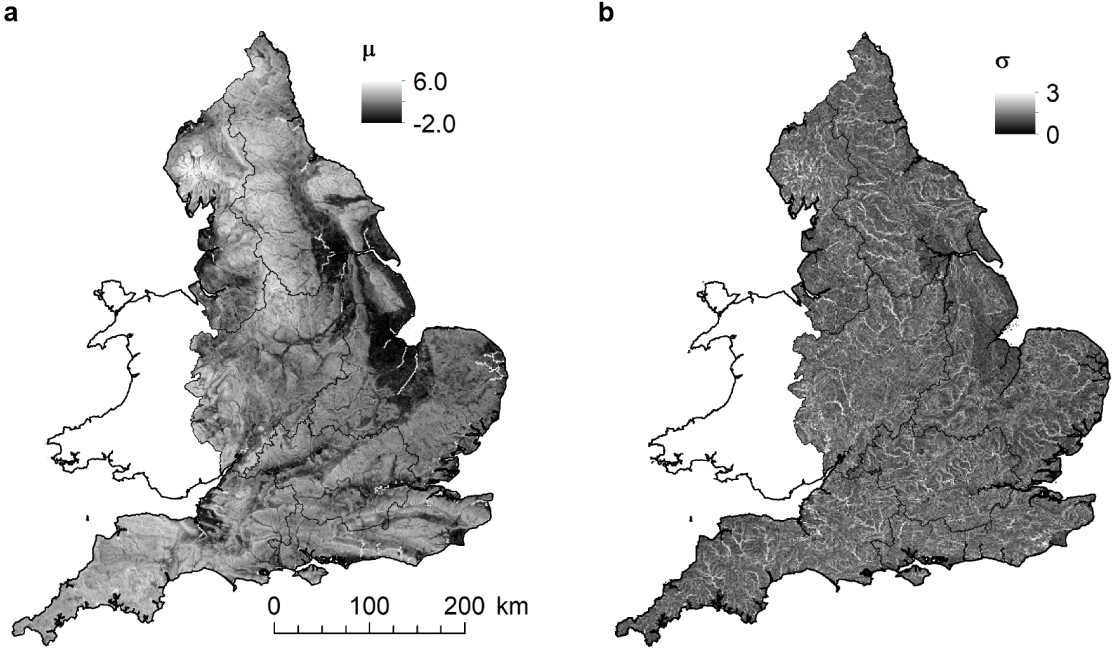

**Figure 7.** Sub-grid inundation model parameters fitted to 50 m IHDTM data using maximum likelihood estimators. a, $\mu$ the log-mean of sub-grid elevation above grid box minimum and b, $\sigma$ standard deviation of the natural logarithm of elevation above grid box minimum.

### 2.4.2 Quantile-based sub-grid floodplain model

The quantile-based approach to discretising the sub-grid floodplain adopted here extends the CaMa-Flood approach of Yamazaki et al. (2011), which has been used extensively at 0.25° resolution globally. In this study we express the sub-grid elevation, $z(x,y)$, above the minimum elevation in the 1 km grid box using $k$ equally-spaced quantiles which are defined using a cumulative distribution derived from the 50 m IHDTM (Figure 6). In the present case, $k = 10$ (i.e., deciles following Yamazaki

10 et al., 2011). Once the depth above bankfull is diagnosed from Equations 3–5, the inundated fraction is obtained as the inverse of the cumulative distribution in Figure 6, interpolated where necessary to ensure conservation of mass. The quantile-based





method is more computationally demanding than the distribution-based method described in Section 2.4. Runtime for this part of the code is 8.51 s with the quantile-based method, which is 238 times slower than the distribution-based method, which takes 0.036 s. Whilst these times are short enough not to impact the viability of either approach here, in future applications with higher resolution, larger domains, or large ensembles, the difference may become of practical importance. The additional

topographic complexity retained in the quantile-based approach increases the size of the supporting datasets by a factor of four, from 7 MB to 28 MB in the present example.

## 2.5 Environment Agency benchmark data

To evaluate our results, we use the latest version of the Indicative Floodplain Map for England (Figure 2a) which was compiled from local assessments of flood risk merged with a national-scale analysis (National Audit Office, 2001). The merged product

delineates flood risk in zones based on estimated return period of inundation. In the present study we evaluate our model results against the indicative area of Flood Zone 3, which corresponds to the area likely to be inundated with AEP of 0.01. We consider only the fluvial area of Flood Zone 3 (i.e., not those parts of the indicative map which are attributed to tidal or coastal influences); nor do we consider areas that are defended against flooding, as designated in the benchmark dataset. The benchmark data were converted from their native vector format to a flooded fraction on a regular grid with 1 km horizontal

resolution identical to our model solution mesh.

Summary statistics for the benchmark data are given in Table 3. Regions used in the present analysis are defined according to the hydrometric regions used in the National River Flow Archive (NRFA) reporting protocols (Marsh and Hannaford, 2008). The Anglian region of England, UK, contains the greatest proportion of its area in Flood Zone 3 (21%), although only half of that area is at risk solely from fluvial inundation. By contrast, only 6% of the Thames region is at risk from flooding although

being furthest from the coast the Thames and the Midlands experience the bulk of their flood hazard (by area) from fluvial flooding (86% and 95% respectively).

## 2.6 Validation metrics

To assess model performance we compute a series of categorical metrics designed to capture successful and unsuccessful outcomes in forecast situations (Mason, 2003; Stephens et al., 2014; Wing et al., 2017). The observed and modelled inundated

fractions, $f_o$ and $f_m$, are defined for each grid box respectively as the ratio of inundated area to total area within the grid box such that $f_o, f_m \in [0, 1]$. We use Boolean metrics $O$ and $M$ for each grid box defined as:

$$
O = \begin{cases} 0 & f_o \leq \varepsilon \\ 1 & f_o > \varepsilon, \end{cases} \tag{14}
$$

$$
M = \begin{cases} 0 & m_o \leq \varepsilon \\ 1 & m_o > \varepsilon, \end{cases} \tag{15}
$$





**Table 3.** Topographic properties by EA region including fraction of region occupied by fluvial and tidal flood risk zones. F&T, fluvial and tidal.

| Region | Total area | Floodplain area | | | | |
| --- | --- | --- | --- | --- | --- | --- |
| | | Fluvial | F&T | Tidal | Total | Fluvial |
| | km$^2$ | km$^2$ | km$^2$ | km$^2$ | % | % |
| Anglian | 26,824 | 2,892 | 1,489 | 1,226 | 20.9 | 52 |
| Midlands | 21,483 | 1,547 | 18 | 61 | 7.6 | 95 |
| North East | 22,929 | 1,307 | 242 | 342 | 8.2 | 69 |
| North West | 14,151 | 628 | 345 | 121 | 7.7 | 57 |
| South West | 20,937 | 900 | 343 | 203 | 6.9 | 62 |
| Southern | 11,054 | 432 | 468 | 185 | 9.8 | 40 |
| Thames | 13,023 | 707 | 99 | 15 | 6.3 | 86 |
| **Total** | **130,401** | **8,413** | **3,004** | **2,153** | **10.4** | **62** |

where $\varepsilon$ is a threshold of detection set in the present case to the precision afforded by the horizontal resolution of the sub-grid data, $\Delta x_s = 50$ m, such that:

$$\varepsilon = \left( \frac{\Delta x_s}{\Delta x} \right)^2 = 0.0025. \tag{16}$$

In order to replicate the benchmark calculation, locations with upstream area $< 10$ km$^2$ are masked (Morris and Flavin, 1990)
5  in this evaluation. Equations 17 and 18 define a hit rate or probability of detection, HR, and a false alarm rate, FAR, which measures the probability of a false positive:

$$\text{Hit Rate, HR} = \frac{\Sigma M \wedge O}{\Sigma O}, \text{ and} \tag{17}$$

$$\text{False Alarm Rate, FAR} = \frac{\Sigma M \wedge \neg O}{\Sigma M}, \tag{18}$$



wherein the standard Boolean conjunction, $\wedge$, disjunction, $\vee$, and complement, $\neg$, operators are used and summation is over all valid members of the sets $O$ and $M$ described above. HR and FAR are probabilities each defined on the real interval [0,1]. We further employ a critical success indicator, CSI, such that:

$$\text{Critical success indicator, CSI } = \frac{\Sigma M \wedge O}{\Sigma M \vee O}, \tag{19}$$

also defined on the real interval [0,1], which rewards successful prediction but penalises false positives; and an error bias metric, EB, defined as:

$$\text{Error bias, EB } = \frac{\Sigma M \wedge \neg O}{\Sigma O \wedge \neg M}, \tag{20}$$

defined on the real interval [0,∞), where values less than unity indicate systematic under-prediction and those greater than unity indicate a tendency to over-predict (Sampson et al., 2015).

## 3  Results

### 3.1  National-scale comparisons

When evaluated across the study region, the log-normal sub-grid floodplain model performs acceptably, but not as well as the quantile-based method. We obtain a hit rate (probability of detection) of 71%, a false alarm rate of 9% and a critical success score of 67% (Table 4). The error bias using this approach was 0.21 indicating that, when in error, the model was five
times more likely to have missed a legitimate flood than to have generated a false alarm. Better results were obtained with the quantile-based floodplain model, which accurately simulates flooding in 95% of grid boxes which experience inundation in the benchmark dataset, with only 10% false positives (Table 5). The critical success indicator, which penalises the model for its errors is therefore 86% for the entire modelled domain. A domain-wide error bias of 2.06 indicates that predictions err on the side of caution. That is, the model is more likely to generate false alarms than it is to miss areas of true inundation in
the benchmark dataset. These performance metrics are comparable with those obtained in previous studies using models with more explicit computational complexity (Sampson et al., 2015; Sosa et al., 2020; Wing et al., 2017) or with similar sub-grid topographic representations (Johnson et al., 2019; Yamazaki et al., 2011).

The spatial pattern of model performance is plotted in Figure 8a,b and summarised in Tables 4 and 5. Areas where the mechanism of flooding is tidal rather than fluvial are masked out, as are areas defended using artificial structures. As is evident
from the model performance statistics, the log-normal model underestimates flood extent, although with no systematic pattern to the discrepancy (Figure 8a). The quantile-based model accurately captures zones of regionally-significant fluvial inundation along major water courses in areas of known flood impact, most notably those in flat terrain of England such as the Somerset



Levels, Fens of East Anglia, and low-lying parts of North Yorkshire (Figure 8a). The Weald, and low-lying coastal regions of Kent underlain by impermeable lithology, and the Severn regions of Shrewsbury and Tewkesbury are accurately simulated in the model. Key locations with important flood impacts around the Trent, and the Yorkshire Ouse, Humber region and the Vale of Pickering are also accurately modelled Figure 8a,b.

5    In general, the study results take account of the important geological differences present across the study region, largely due to the use of information from the HOST classification in direct flood depth estimation. Igneous and metamorphic rocks found predominantly in the north and west of the region contrast notably with highly permeable chalk in the south and east. The latter formation, a thickly bedded Upper Cretaceous limestone found in Hampshire, the Chilterns, North and South Downs, East Anglia and the Lincolnshire and Yorkshire Wolds is broadly accounted for in the flood depth estimation process. Nonetheless,

10   there remain a significant number of false alarms particularly in the headwaters of rivers in catchments underlain with chalk. We comment on this behaviour and possible remedies in the next section.

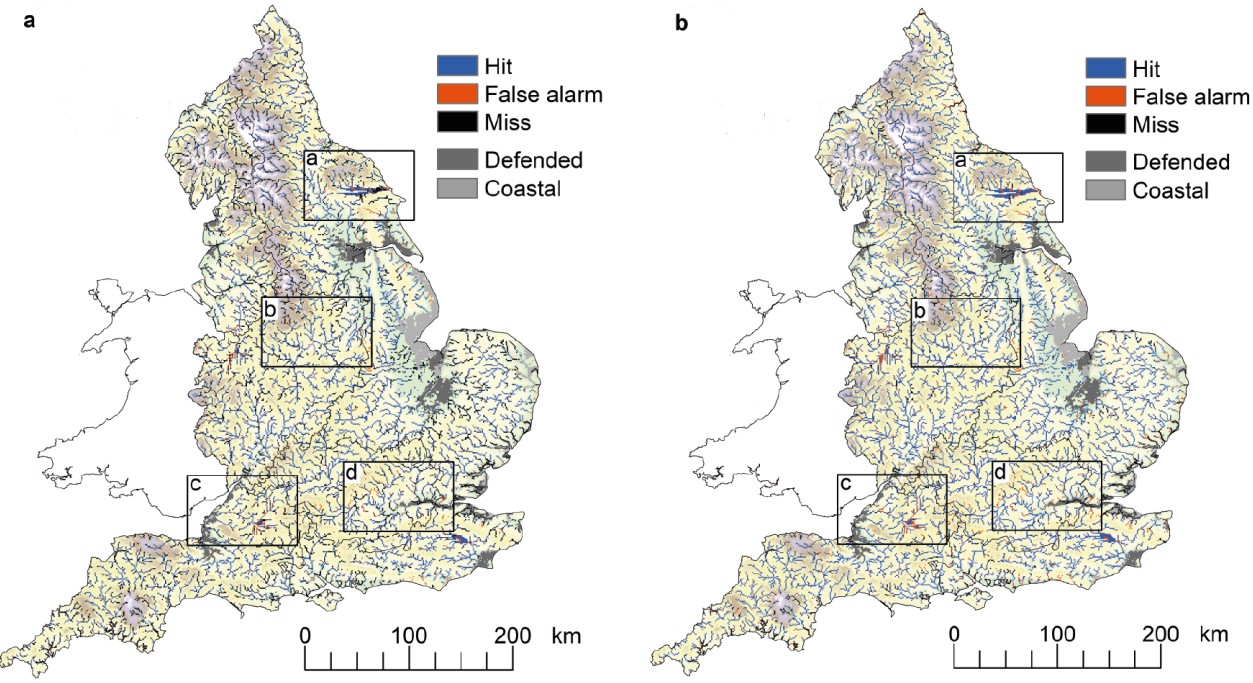

**Figure 8.** Model performance indicators: blue, hit; red, false alarm; black, miss; dark grey, defended areas; mid-grey, areas of coastal flooding for a, log-normal distribution-based sub-grid floodplain model; b, quantile-based sub-grid floodplain model. Boxes labelled a–d correspond to subpanels plotted in Figure 9.





## 3.2  Regional evaluation

Regional performance metrics are listed in Table 4 and 5. Patterns of model performance are broadly consistent between the quantile-based and the log-normal distribution-based models, though the latter is systematically better. For the log-normal distribution-based model, the critical success measure ranges between 59% and 71%, with the Thames and Midlands seeing the

best probability of detection (hit rates 75% in each case; Table 4). The highest false alarm ratio (12%) for this floodplain sub-model is seen in the South East. For the quantile-based model, performance is best in the Midlands and East Anglia, followed by the North East and North West. In the Thames the hit rate is highest of all locations (99%) but performance measured using the critical success index is reduced slightly by the relatively high rate of false positives (11%) compared with other regions, which is due to the dominance of chalk. The model performs least well in the South East and South West although in all regions

more than 80% of the inundation in the benchmark dataset is accurately simulated. In the former case this is almost certainly due to the influence of highly-permeable chalk in Hampshire, the Chilterns and the North and South Downs. By contrast, the error bias is lowest (although still above unity) in the South West and North West regions because there are areas of missed inundation on shallow, upland organic soils.

**Table 4.** Performance indicators for log-normal floodplain model broken down by EA reporting regions. See text for definitions of regions and metrics.

|  | Hit rate | False alarm ratio | Critical success index | Error bias |
|---|---|---|---|---|
| **North East** | 0.72 | 0.09 | 0.67 | 0.25 |
| **East Anglia** | 0.71 | 0.07 | 0.68 | 0.17 |
| **South West** | 0.67 | 0.09 | 0.63 | 0.20 |
| **South East** | 0.71 | 0.12 | 0.65 | 0.33 |
| **North West** | 0.62 | 0.07 | 0.59 | 0.13 |
| **Midlands** | 0.75 | 0.07 | 0.71 | 0.21 |
| **Thames** | 0.75 | 0.10 | 0.69 | 0.34 |

Detailed spatial maps showing model performance in key regions are shown in Figure 9. For brevity, only results for the

quantile-based model are plotted here. The Vale of Pickering and tributaries of the Derwent to the north-east of York in North Yorkshire are shown in Figure 9a. In this location, the broad patterns of large-scale flood inundation are captured well, even north into the North York Moors, although some isolated grid boxes do not register flooding in locations where thin upland soils do, in fact, produce inundation. By contrast, to the south and east of the area the model raises several false alarms which are associated with a lower level of flood risk. In common with other areas underlain with permeable chalk, parts of the Yorkshire

Wolds are also troubled by some consistent false alarms.





**Table 5.** Performance indicators for quantile-based floodplain model broken down by EA reporting regions. See text for definitions of regions and metrics.

|  | Hit rate | False alarm ratio | Critical success index | Error bias |
|---|---|---|---|---|
| **North East** | 0.98 | 0.08 | 0.90 | 4.13 |
| **East Anglia** | 0.98 | 0.07 | 0.91 | 4.06 |
| **South West** | 0.94 | 0.09 | 0.86 | 1.60 |
| **South East** | 0.94 | 0.14 | 0.81 | 2.58 |
| **North West** | 0.96 | 0.07 | 0.90 | 1.49 |
| **Midlands** | 0.98 | 0.06 | 0.92 | 3.49 |
| **Thames** | 0.99 | 0.11 | 0.89 | 9.70 |

The Midland region plotted in Figure 9b, which is centred on the River Trent, is relatively well simulated with few false alarms. An isolated number of grid boxes which register as 'misses' are located in the north of the region, in the thinner soils of the Peak District. Figure 9c shows a region of Somerset and Wiltshire in which model performance is generally sound but with a tendency towards false alarms, particularly in areas underlain by Jurassic clay. The part of the Thames basin shown in Figure 9d is, in general, well simulated, notwithstanding the false alarms associated with direct flood depth estimation in the chalk of the Chilterns and the North Downs, which have been discussed previously. The presence of such false alarms is unlikely to impede the interpretation of model outputs in practice because they occur in regions where hydrological behaviour is known to be controlled strongly by geology.



**Figure 9.** Regional model performance assessment centred on a, Vale of Pickering; b, Derbyshire Trent; c, Somerset-Wiltshire; d, Greater London Thames; see Figure 8 for definition of shading. Yellow zone masks area protected by flood defences. Benchmark Flood Zone data contain public sector information licensed under the Open Government Licence v3.0. Base maps contain Ordnance Survey data © Crown Copyright and database rights 2019





## 4 Discussion

### 4.1 Sensitivity analysis

The two steps in the process chain with the potential to contain the highest unconstrained uncertainties are the fitting of $h_{100}$ and the specification of frictional roughness via the Gauckler-Manning coefficient, $n$. Below we tabulate the sensitivity of
5 model performance to these parameters (Tables 6 and 7).

**Table 6.** Sensitivity to parameters $m$ and $n$ for the log-normal floodplain model.

|  | Hit rate | False alarm ratio | Critical success index | Error bias |
|---|---|---|---|---|
| **Growth factor, $m$** | | | | |
| -20% | 0.62 | 0.09 | 0.59 | 0.13 |
| **Baseline** | 0.71 | 0.09 | 0.67 | 0.21 |
| +20% | 0.77 | 0.09 | 0.72 | 0.31 |
| **Gauckler-Manning coefficient, $n$** | | | | |
| ×0.5 | 0.74 | 0.09 | 0.67 | 0.24 |
| **Baseline** | 0.71 | 0.09 | 0.67 | 0.21 |
| ×2.0 | 0.69 | 0.09 | 0.65 | 0.21 |

**Table 7.** Sensitivity to parameters $m$ and $n$ for the quantile-based floodplain model.

|  | Hit rate | False alarm ratio | Critical success index | Error bias |
|---|---|---|---|---|
| **Growth factor, $m$** | | | | |
| -20% | 0.90 | 0.10 | 0.82 | 0.93 |
| **Baseline** | 0.95 | 0.10 | 0.86 | 2.06 |
| +20% | 0.97 | 0.10 | 0.87 | 3.23 |
| **Gauckler-Manning coefficient, $n$** | | | | |
| ×0.5 | 0.96 | 0.10 | 0.87 | 2.62 |
| **Baseline** | 0.95 | 0.10 | 0.86 | 2.06 |
| ×2.0 | 0.94 | 0.10 | 0.85 | 1.52 |

The growth factor, $m$, used to estimate $h_{100}$ is altered from 80% to 120% of the fitted value. This change results in only a small change in overall model performance (Table 6). Specifically, a 20% reduction in $m$ (and therefore in $h_{100}$) leads to





a lower hit rate together with a lower false alarm rate which together reduce the error bias accordingly. However, the critical success index shows that the reduction in hit rate is not fully compensated for by the reduced occurrence of false alarms. Conversely, a 20% increase in $h_{100}$ appears to have little impact on model performance other than to increase the error bias slightly. The model is largely insensitive to the choice of the Gauckler-Manning coefficient, $n$, with the only discernible effect
being a slightly lower error bias with higher values of $n$.

## 4.2   Applications

The central aim of this paper has been to demonstrate that combining a 1 km inundation model based on a simplified inertial form of the shallow water equations with a sub-grid representation of floodplain topography can produce acceptable wide-area flood simulations. In relation to our first hypothesis, we conclude that it is possible to achieve acceptable performance against
an established benchmark at coarse resolution using a fast inundation algorithm with a sub-grid flood plain. Moreover, in relation to our second hypothesis, we conclude that using a quantile-based sub-grid floodplain model increases the probability of detection by 20% whilst increasing the likelihood of a false alarm by only 1%.

The computational efficiency of our approach — achieved through sub-grid parametrisation — is intended to benefit applications where rapid assessments are required for early-look warnings prior to hydrodynamic analysis with finer precision.
The additional performance of the quantile-based model comes at the expense of a 4-fold increase in the size of the supporting sub-grid topographic datasets, and a 238-fold increase in computation time. In spite of this extra expense, which we note is still small compared with the additional cost of running an inundation model at the native 50 m sub-grid scale, it is possible to compute the steady-state AEP 0.01 flood extent for the British mainland in approximately ten minutes. Moreover, the ability to produce many rapid simulations with only modest computational resource opens possibilities either for higher-resolution
applications or wider-ranging quantification of uncertainties (Hrachowitz and Clark, 2017). In locations where flood discharge or level data are unavoidably imprecise, the potential to provide fast, accurate simulations to partition uncertainties between initial conditions, driving data and topographic boundary conditions – and to test hypotheses about model structure – may also be beneficial (Beven et al., 2020).

## 4.3   Future Work

The method used here can be applied regionally or globally and extended to use other datasets at finer resolution. As indicated in the introduction, this study represents a first step in providing a component for integration into the Joint UK Land Environment Simulator. Further work is underway to make this coupling, which will enable the community to simulate the wider impacts of flood inundation on the Earth system including its links with: (i) surface energy exchange (Dadson et al., 2010; Decharme et al., 2008), (ii) focused groundwater recharge (Taylor et al., 2013), (iii) carbon-cycle biogeochemistry of vegetation (Gedney
et al., 2004), and (iv) shelf-seas and the coastal ocean as part of a coupled system (Lewis et al., 2018). Additional work is planned to evaluate the model's applicability for wide-area flood forecasting and climate impact assessments and to assess its performance during specific, high-impact flood events. We also note the potential to impose a time-varying lower (coastal) boundary condition either to simulate coastal flooding as a distinct process or alongside coincident fluvial flooding.





## 5 Conclusions

This study has sought to introduce and evaluate an efficient approach that can deliver fast, accurate predictions of fluvial inundation over a wide area. We have combined a solution to the inertial form of the shallow water equations with a sub-grid representation of floodplain topography and have driven it with flood depth estimates from an established procedure used to
5    define the AEP 0.01 flood. When evaluated against Environment Agency benchmark data, this approach produces accurate estimates of flood inundation in 86% of locations with a domain-wide hit rate of 95% and a false alarm rate of 10%.

Our evaluation highlights the need for accurate initial conditions and input data, and we propose further work to validate the model for the transient case and for cases where the downstream boundary is controlled by coastal influences. Our approach is computationally expedient and permits both large-area real-time forecasts of flood hazard at national scale whilst also allowing
10    fully-coupled Earth system models to represent explicitly the links between inundated extent and processes in the overlying atmosphere.

*Data availability.* Supporting datasets and model results necessary to reproduce the figures shown in this study are available for public download from the NERC Environmental Informatics Data Centre (https://doi.org/10.5285/tbc). All model results can be reproduced using the equations given in the text.

15 ## Appendix A: Definitions

Table A1: Notation used for quantities defined in this paper

| Notation | Quantity | Dimensions | Unit |
|---|---|---|---|
| $A_{bf}$ | Channel cross-sectional area | $[L^2]$ | $m^2$ |
| $w_{bf}$ | Bankfull channel width | $[L]$ | m |
| $h$ | Inundated depth | $[L]$ | m |
| $h_{bf}, h_{100}$ | Bankfull depth; AEP 0.01 depth | $[L]$ | m |
| $\Delta x$ | Horizontal grid box length | $[L]$ | m |
| $A_{fl}$ | Inundated fraction | $[-]$ | $[-]$ |
| $V$ | Prognostic water volume in grid box | $[L^3]$ | $m^3$ |
| $A_{cat}$ | Catchment area | $[L^2]$ | $km^2$ |
| SAAR | Standard annual average rainfall | $[L]$ | mm |
| SPRHOST | Standardised percentage runoff from HOST classification | $[-]$ | % |
| $a,b,c,d$ | Constants in estimator for $h_{bf}$ | See text | See text |
| $m,p$ | Constants in growth curve to estimate in 0.01 AEP flood depth from $h_{bf}$ | See text | See text |



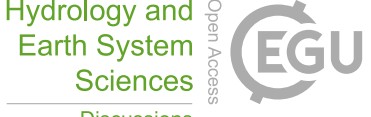

Continuation of Table A1

| Notation | Quantity | Dimensions | Unit |
|---|---|---|---|
| $q$ | Lateral flow per unit channel width between grid boxes | $[L^2T^{-1}]$ | m$^2$ s$^{-1}$ |
| $g$ | Gravitational acceleration | $[LT^{-2}]$ | m s$^{-2}$ |
| $h_t$ | Water depth (at time $t$) | $[L]$ | m |
| $z$ | Topographic elevation | $[L]$ | m |
| $t$ | Time co-ordinate | $[T]$ | s |
| $x_i$ | Space co-ordinates | $[L]$ | m |
| $n$ | Gauckler-Manning coefficient | $[TL^{-1/3}]$ | s m$^{-1/3}$ |
| $\Delta t$ | Time step | $[T]$ | s |
| $h_{max}$ | Maximum water elevation across domain | $[L]$ | m |
| $R$ | Hydraulic radius | $[L]$ | m |
| $\alpha$ | Numerical stability factor | $[-]$ | $[-]$ |
| $\tau^*$ | Time-constant for model equilibration | $[T]$ | s |
| $A_{cat}^*$ | Maximum inundated area | $[L^2]$ | km$^2$ |
| $c^*$ | Characteristic wave celerity | $[LT^{-1}]$ | m s$^{-1}$ |
| $\bar{h}$ | Typical water elevation across domain, used in time-constant calculation | $[L]$ | m |
| $\bar{S}$ | Typical channel slope used in calculation of time constant | $[-]$ | $[-]$ |
| $A(z^*)$ | Function describing area fraction of grid box below a given elevation, $z^*$ | $[-]$ | $[-]$ |
| $s, \xi$ | Argument and integration variable for definition of error function in Equation 13, respectively | $[-]$ | $[-]$ |
| $k$ | Number of quantiles used to represent sub-grid floodplain | $[-]$ | $[-]$ |
| $f_o, f_m$ | Inundated fraction, observed and modelled respectively | $[-]$ | $[-]$ |
| $O, M$ | Boolean indicators of inundation fraction greater than $\varepsilon$ for observed and modelled inundation, respectively | $[-]$ | $[-]$ |
| $\varepsilon$ | Threshold of detection | $[-]$ | $[-]$ |
| $\Delta x_s$ | Horizontal resolution of sub-grid data | $[L]$ | m |




*Author contributions.* SD initiated and led the study. SD wrote the model and produced the results with the following contributions: (i) HD produced the topographic ancillary files and data on channel width; (ii) RP derived the roughness map; (iii) TM and HL integrated a prototype of the model into JULES. All co-authors contributed to the design of the study. The manuscript was prepared by SD with contributions from all co-authors.

5 *Competing interests.* The authors declare no competing interests.

*Acknowledgements.* We thank Paul Bates, Jeff Neal and Dai Yamazaki for discussions in advance of this study which illuminated the possible pathways ahead. Environment Agency Flood Risk benchmark data were obtained under Open Government Licence 3.0 from https://environment.data.gov.uk/. We acknowledge funding for this work from the UK Natural Environment Research Council (NE/S017380/1 and NE/I01277X/1).





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
