# Peer review of "A reduced-complexity model of fluvial inundation with a sub-grid representation of floodplain topography evaluated for England, United Kingdom"

_Hydrology and Earth System Sciences, 2021_

## Referee Comment (RC2)

The paper is well written, covers a large literature review in the hydrology field, but according to me completely fails in providing a convincing motivation of the hydrodynamic structure of the proposed model. More specifically:

1) It is not clear if the model is a 1D or a 2D model. Eq. (3) is the continuity equation of a 1D model, where the flow along the direction normal to direction xi is zero. The same holds for the momentum equation (4). If Eqs (3) and (4) hold for direction x1, they cannot hold for direction x2. On the other hand, authors adopt a regular structured grid, with grid size of 1 km.

2) I assume the water depth is updated at the new time level from the finite difference approximation of the continuity equation (3), but this is not discussed in the paper.

3) If Eq. (4) is the momentum equation along the xi direction of a 2D model, the 3$^{rd}$ resistance term on its l.h.s. must be written in the form:

$$\frac{gn^2 q_i \sqrt{q_i^2 + q_j^2}}{h^{7/3}} \, ,$$

otherwise it depends on the grid orientation. If Eq. (4) is the momentum equation of a 1D model, the approximation of the hydraulic radius with the hydraulic depth is a very strong one, also because the channel width is a very arbitrary choice.

4) The choice of a zero convective inertia model should be discussed against other possible approximations. It is known that the error of the zero convective inertia model is larger than the error of the zero model ([1]-[4]). A trivial example is the front of a sharp shock wave, where the local inertia is positive, but the convective inertia is negative. In this case to neglect only one of the two components is worse than to neglect both.
The advantage of the zero inertia model is that it allows an easy solution in the case of small water depths, but there are also other options that can be applied for fully diffusive models ([5]-[6]).

5) The authors carry on a sensitivity analysis of the model results for the choice of the Manning coefficient and of the channel width, but they should do the same also for the topographic elevation z. Because the adopted value, in each computational cell, is the mean elevation computed over a 1kmq area, I assume that both the averaging technique and the measurement error lead to a very large uncertainty.

1) Perumal, M., and K. G. Ranga Raju (1998a), Variable-parameter stagehydrograph routing method. I: Theory, J. Hydrol. Eng., 3(2), 109 – 114.
2) Perumal, M., and K. G. Ranga Raju (1998b), Variable-parameter stagehydrograph routing method. II: Evaluation, J. Hydrol. Eng., 3(2), 115 – 121.
3) Ponce VM, Li RM, Simons DB. Applicability of kinematic and diffusion models. J Hydraul Div, ASCE 1978;104:353–60.
4) Ponce VM. Generalized diffusive wave equation with inertial effects. Water Resour Res 1990;26:1099–101.

5) Sinagra, M., Nasello, C., Tucciarelli, T., Barbetta, S., Massari, C., Moramarco, T. A self-contained and automated method for flood hazard maps prediction in urban areas (2020) Water (Switzerland), 12 (5), art. no. 1266.

6) Aricò, C., Filianoti, P., Sinagra, M., Tucciarelli, T. The FLO diffusive 1D-2D model for simulation of river flooding (2016) Water (Switzerland), 8 (5), art. no. 200.

---

## Community Comment (CC1)

**Unsolicited comment on Dadson et al.:**

***A reduced-complexity model of fluvial inundation with a sub-grid representation of floodplain topography evaluated for England, United Kingdom***

The Bates & Neal Flood Lab, part of the Hydrology Group in the School of Geographical Sciences at the University of Bristol, reviewed this HESS Discussion paper during one of our meetings and provide the following comments that we hope are useful to the authors.

*General comments*

The general framing of the paper does not justify what place a model of this fidelity has in a country like the UK. Where metric-resolution inundation models with gauge-based flows, better parameterised channels, lidar terrain, and flood defences are already available, what is the need for a steady-state, 1 km, undefended model? Observations of flow, channel properties, elevation, and flood defences that are more accurate than the components used here are readily available for the UK.

The justification that simulations are quick does not outweigh the need for accurate models. The authors fail to discuss the merits of sampling from pre-simulated libraries of more accurate flood inundation maps when time is at a premium, for instance, or downscaling the 1 km model back to the native resolution of the DEM.

The other justification that simplified models are yet to be evaluated fully is not the case. There is already a wealth of literature on the general inability of coarse and/or physics-lite models to replicate detailed validation data, some of which the authors themselves cite.

More specific concerns follow.

*Channel and boundary condition configuration*

We were unable to understand the treatment of channels in the model. In particular, whether estimated bankfull depths are "burned" into the DEM or retained subgrid.

It appears that all channel variables are based upon a linear regression of ~30 observations collected 30 years ago. We question how representative such poorly constrained equations are for applications at national scales, particularly since there is no consideration of slope or discharge (the ultimate determinants of hydraulic geometry).

The extreme boundary condition is again based on a simple uplift of the same measurements from the 1991 paper, rather than any understanding of growth curves that the authors' own organisation sets out in the Flood Estimation Handbook. It is also unclear how this boundary condition is input to the model: by being steady-state, the model would struggle to simulate non-valley filling floods.

In a country as data-rich as the UK, there is little need to estimate these properties in the way the authors describe. The use of its rich network of river gauges and channel approximations

based on discharge and slope would undoubtedly be a more justifiable approach than the one taken: it would not "introduce additional uncertainty" (P3/L15), it would decrease it.

If the model could readily receive flows as input, as seems to be suggested through its intended coupling to JULES, how then would channels be parameterised?

*Model validation*

The model validation is questionable. Scaling up the high-resolution benchmark data to that of the coarse model is not a fair test of its skill – the appropriateness of low model resolution is partly what should be tested. Most channels and floodplains in the UK are <1 km wide, meaning (as is shown in Figure 9) the validation procedure simply discriminates whether channels exist in broadly the correct locations and whether water is input to them. To suggest that "these performance metrics are comparable with those obtained in previous studies" is disingenuous when such exercises exhausted the utility of the validation data rather than degrading it to fit the model. For the high-level conclusion to be 86% similarity to EA maps is patently false.

The translation of inundated cell fractions to binary wet/dry grids with a very low threshold of detection ($\varepsilon$) is, again, a very forgiving comparison. It is not clear why the flood fractions are not just compared directly. A more useful test, however, would be to use the validation data at their native resolution.

Oliver Wing

Paul Bates

Jeff Neal

Chris Sampson

Gemma Coxon